# Essential Oil Content and Composition of the Chamomile Inflorescences (*Matricaria recutita* L.) Belonging to Central Albania

**Ivan Salamon** [1],*[ID], **Alban Ibraliu** [2] **and Maryna Kryvtsova** [3]

[1] Department of Ecology, Faculty of Humanities and Natural Science, University of Presov, 01, 17th November St., SK-081 16 Presov, Slovakia

[2] Department of Agriculture Sciences, Faculty of Agriculture and Environment, Agricultural University of Tirana, 1029, Koder Kamez, AL-1021 Tirana, Albania

[3] Department of Genetics, Plant Physiology and Microbiology, Biological Faculty, Uzhhorod National University, 32, Voloshyn St., UA-88000 Uzhhorod, Ukraine

\* Correspondence: ivan.salamon@unipo.sk; Tel.: +421-917984060

**Abstract:** The chamomile drug (*Chamomillae anthodium*) is widely known and has been used as a medicine for ages. Today, the drug is officially registered in the European Pharmacopoeia. Despite the economic importance of the chamomile *Matricaria recutita* L., little is known about the extent and nature of the essential oil variability and composition of this species in Albania. Therefore, information about the extent of the uses of various gene pools is extremely valuable for the development of future chamomile cultivation and breeding programs. This study aimed to analyze the differences among 29 populations in different sites in central Albania. The quantities of essential oils ranged from a low of 0.10 ± 0.01% in Fier and Tirana to a high of 0.75 ± 0.05% in Linzë. The yield of volatile oils depended on the geography, altitude, and other factors, including stress influences on the site of plant population growth. This fact was confirmed by various subclimatic characteristics obtained from individual localities. Essential oil extracted from chamomile inflorescences was recorded to have between 23 and 43 chemical components. It was found that /-/-α-bisabololoxides B and A were the major constituents in 25 samples, with only 4 having dominant /-/-α-bisabolol. The uniquely determined chemical type of the chamomile wild populations in Albania was chemical type B (/-/-α-bisabololoxide B > /-/-α-bisabololoxide A > /-/-α-bisabolol). Based on the study of chamomile's pharmacodynamic properties, the sesquiterpenes /-/-α-bisabolol and chamazulene are considered to be the most valuable constituents. Consequently, a very intensive improvement breeding program must begin, with emphasis on drug yield, polyploidization, essential oil quantity, and both component contents in the oil.

**Keywords:** chamomile; chemical type; essential oil; GC-MS; sesquiterpenes; central Albania





## 1. Introduction

Chamomile, *Matricaria recutita* L., of the *Asteraceae* family, is of great importance as a raw material in pharmaceutical, cosmetic, and food industries [1]. The species is native to eastern and southern Europe as well as parts of western Asia and now occurs throughout almost all of Europe as well as Turkey and the Caucasus region (Georgia), in addition to parts of Iran and Afghanistan [2]. The material of commerce is obtained, for the most part, from farms in Egypt, Germany, Argentina, Poland, and, to a lesser extent, Chile [3], the Czech Republic, Slovakia, Spain, and several of the Balkan countries (Bosnia and Herzegovina, Bulgaria, Croatia, and Serbia). It is also wild-collected for commercial trade in Hungary, as well as in Albania, Bulgaria, Croatia, Kosovo, and North Macedonia [4,5].

In Albania, chamomile plants are commonly consumed as a tea [6]. They have a long history as an effective herbal remedy used in folk medicine to treat fevers, common colds, cough stomach ailments, and gastrointestinal disorders, and their constituents show

antiseptic, anti-inflammatory, and antispasmodic properties. They are usually drunk as an infusion but can also be added to bathwater to soothe dermatitis. The chamomile drug in pharmaceutic products is commonly used in the form of macerates, extracts, infusions, and essential oils. Similar raw materials are added to a variety of cosmetic preparations such as creams, soaps, skin lotions, bath ingredients, hair shampoos, etc. [7].

The geographical distribution of ubiquitous chamomile populations in the Balkan country involves mainly grassy sites, uncultivated areas, along roads, and in house yards; they are less cultivated as a commercial medicinal plant. They reach 1200 m in altitude, ranging from fields to alpine valleys. They occur in sun-loving (heliophilous) or medium shade-loving (mesoskiophilous) conditions. The species is commonly found in rich base soils, more or less rich in nutrients with a basic or slightly acidic pH, and very dry and cool soils [8].

This complex study presents the yield variation of chamomile essential oils and the sesquiterpene compositions (/-/-α-bisabololoxide B, /-/-α-bisabololoxide A, /-/-α-bisabolol, chamazulene, and en-yn-dicycloethers) of autochthonous populations in central Albania. Despite the economic importance of the chamomile, *Matricaria recutita* L., little is known about the extent and nature of the essential oil variability and composition of this species in Albania. Therefore, information about the extent of uses of various gene pools is extremely valuable for the development of future chamomile cultivation and breeding programs.

## 2. Material and Methods

### 2.1. Collection of the Plant Material

Inflorescences were collected from 29 localities in central Albania in period from 2017 to 2019 (Table 1, Figure 1). The typical biotopes and plant formations of this plant species were dry lawns, forest margins, footpath margins, house yards, countryside, and around ruins. It is principally associated with forest margins (*Origanetalia*) and invasive oak forest (*Quercus pubescenti-petraeae*) [9].

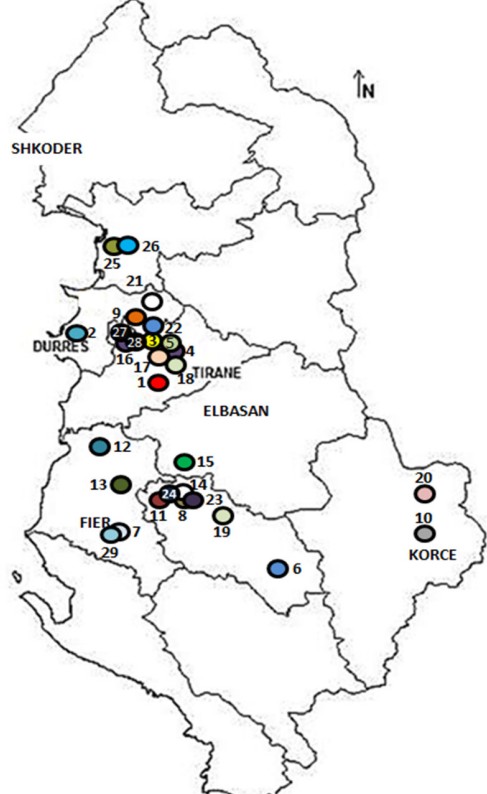

**Figure 1.** Albania (www.d-maps.com) with its prefectures and the localities, where chamomile flower samples were collected: 1. Baldushk, 2. Rruga Gjiri i Lazit, 3. Zall Her, 4. Dajti, 5. Linzë, 6. Skrapar: Melove, 7. Verbas, 8. Kuçovë, 9. Krujë: Larushk, 10. Korçë, 11. Poshnje: Berat, 12. Karbunarë Lushnjë,

13. Karbunarë e Vogël, 14. Kuçovë: Rruga S. Zuber, 15. Kuçovë: Rruga Kozare, 16. Gjokaj, 17. Tirana: Botanic Garden, 18. Tirana: Grand Park, 19. Skrapar: Tomorri Mount., 20. Sheqeras, 21. Krujë: Droja, 22. Tapizë, 23. Kuçovë: Parku i Qytetit, 24. Poshnje, 25. Lezhë, 26. Lezhë: Kolsh, 27. Fushë Prezë: Palaq, 28. Fushë Prezë, 29. Fier.

**Table 1.** The basic characters of selected sites in Albania with the occurrence of chamomile populations.

| | Locality | Geographical Latitude | Terrestrial Longitude | Altitude [m] | Aspect | Slope |
|---|---|---|---|---|---|---|
| 01 | Baldushk | N 40°38′43″ | E 20°59′38″ | 250 | south-west | 30° |
| 02 | Rruga Gjiri i Lazit | N 41°37′17″ | E 19°60′56″ | 27 | north-west | 10° |
| 03 | Zall Her | N 41°38′92″ | E 19°82′59″ | 115 | north-west | 0° |
| 04 | Dajti | N 41°21′57″ | E 19°55′32″ | 450 | north-west | 15° |
| 05 | Linzë | N 41°35′34″ | E 19°88′35″ | 400 | north-west | 12° |
| 06 | Skrapar: Melove | N 40°58′85″ | E 20°28′75″ | 879 | South | 0° |
| 07 | Verbas | N 40°70′16″ | E 19°61′59″ | 22 | south-west | 7° |
| 08 | Kuçovë | N 40°80′44″ | E 19°91′34″ | 44 | south-west | 5° |
| 09 | Krujë: Larushk | N 41°43′68″ | E 19°73′27″ | 33 | north-west | 17° |
| 10 | Korçë | N 40°61′58″ | E 20°77′72″ | 870 | south-east | 27° |
| 11 | Poshnje Berat | N 40°78′29″ | E 19°84′43″ | 26 | south-west | 30° |
| 12 | Karbunarë Lushnjë | N 40°92′60″ | E 19°76′82″ | 99 | south-west | 0° |
| 13 | Karbunarë e Vogël | N 40°91′92″ | E 19°71′89″ | 16 | south-west | 0° |
| 14 | Kuçovë: Rruga S. Zuber | N 40°80′13″ | E 19°91′49″ | 48 | south-west | 16° |
| 15 | Kuçovë: Rruga Kozare | N 40°86′78″ | E 19°92′10″ | 37 | south-west | 2° |
| 16 | Gjokaj | N 41°35′87″ | E 19°67′38″ | 88 | north-west | 1° |
| 17 | Tirana: Botanic Garden | N 41°32′68″ | E 19°81′87″ | 200 | north-west | 13° |
| 18 | Tirana: Grand Park | N 41°31′14″ | E 19°82′94″ | 154 | north-west | 21° |
| 19 | Skrapar: Tomorri Mount. | N 40°66′12″ | E 20°17′01″ | 1708 | South | 30° |
| 20 | Sheqeras | N 40°74′53″ | E 20°77′07″ | 815 | south-east | 30° |
| 21 | Krujë: Droja | N 41°51′11″ | E 19°79′23″ | 543 | north-west | 30° |
| 22 | Tapizë | N 41°41′61″ | E 19°75′99″ | 55 | north-west | 31° |
| 23 | Kuçovë: Parku i Qytetit | N 40°80′45″ | E 19°91′44″ | 46 | south-west | 10° |
| 24 | Poshnje | N 40°78′29″ | E 19°84′43″ | 26 | south-west | 30° |
| 25 | Lezhë | N 41°77′65″ | E 19°64′74″ | 0 | north-west | 8° |
| 26 | Lezhë Kolsh | N 41°77′87″ | E 19°68′85″ | 336 | north-west | 30° |
| 27 | Fushë Prezë: Palaq | N 41°40′24″ | E 19°66′92″ | 52 | north-west | 9° |
| 28 | Fushë Prezë | N 41°40′24″ | E 19°66′92″ | 52 | north-west | 4° |
| 29 | Fier | N 40°72′51″ | E 19°55′82″ | 20 | south-east | 0° |

The flower heads were separated and dried in a sheltered, open air area at a temperature below 32 °C for 14 to 20 days with low humidity of 2 to 5%. The moisture content of the flower tissue was lowered to 12% to prevent mold infection. The dried materials were cleaned, packed, labelled, and stored in a clean and dry place until further use.

## 2.2. Chamomile Oil Isolation

Each sample of dried flower with a weight of 20 g was ground in a blender. The essential oil from this raw material was prepared using hydro-distillation (2 h) in Clevenger-type apparatus according to the European Pharmacopoeia (Ph. Eur.) 10th Edition [10]. Hexane was used as a collecting solvent. The essential oils were stored under $N_2$ at +4 °C in a dark space before their GC-FID and GC-MS analyses.

## 2.3. GC-FID Analysis

The analysis of the chamomile essential oils was carried out using a Vega Series Carloerba gas chromatograph, connected to a Spectrophysics SP 4270 integrator. The following operating conditions were used: column: DB5, 30 m × 0.32 mm inner diameter (i.d.); film thickness: 0.25 mm; carrier gas: nitrogen, adjusted to a flux of 1 mL per min; and injection and FID-detector temperatures: 220 and 250 °C, respectively. Components were identified using their GC retention times, and the resulting values were comparable to those of the literature. Oil component standards for comparison were supplied by Extrasynthese, s.a., Genay, France.

## 2.4. GC/MS Analysis

GC/MS analysis was carried out on a Varian 450-GC connected with a Varian 220-MS. The separation was achieved using a Factor Four TM: Capillary Column VF 5 ms (30 m × 0.25 mm i.d., 0.25 μm film thickness). Injector type 1177 was heated at temperature 220 °C. Injection mode was split less (1 μL of a 1:1000 n-hexane/diethyl ether solution). Helium was used as a carrier gas at a constant column flow rate of 1.2 mL per min. Column temperature was programmed: initial temperature was 50 °C for 10 min, then up to 100 °C at 3 °C per min, isothermal condition for 5 min and then continued to heating at 150 °C at 10 °C per min. Total time for analysis of one sample was completed in 59.97 min. Identification of components was performed by comparison of their mass spectra with those stored in NIST 02 mass spectra from the literature [11] and a home-made library, as well as on comparison of their retention indices with the standards.

## 2.5. Statistical Analysis

Statistical analysis was performed using confidence intervals with a significance level $p < 0.05$ using calculation through the mean, standard deviation, and standard error.

The differences between chamomile populations for the mean values of the essential oil constituents were carried out using ANOVA analysis. All statistics data were calculated employing the SAS JMP Statistical Discovery and a dendrogram (ward method) [12], and a relationship diagram (chamomile populations x the essential oil constituents) was carried out.

## 3. Results

### 3.1. The Dependence of Essential Chamomile Oil Yield on Their Habitats

The chamomile essential oil hydro-distilled from the fresh dried flower samples from all localities was pastel bluish and later turned greenish yellow. The oil from the Albanian locations of Zall Her (locality 03), Dajti (locality 04), Kuçovë (locality 08), and Linzë (locality 05) had a deeper blue color. The yield ranged from $0.10 \pm 0.01\%$ (*v/w*) in Fier (locality 29) and the Botanical Garden in Tirana (locality 17) to $0.75 \pm 0.05\%$ (*v/w*) on a dry-weight basis in Linzë (locality 05) (Table 2). The quantity of extractible substances using 60% ethanol did not correspond to the amount of essential oil isolated from the flower heads (Table 2).

**Table 2.** Chamomile inflorescence's yield of essential oil in % (*v/w* – expressed as a dry weight).

| | Habitat/Locality | Extractible Substances using 60% Ethanol [%] | Essential Oil Yield [%] |
|---|---|---|---|
| 01 | Baldushk | 24.0 ± 0.5 | 0.40 ± 0.05 |
| 02 | Rruga Gjiri i Lazit | 23.5 ± 0.5 | 0.40 ± 0.05 |
| 03 | Zall Her | 33.0 ± 0.5 | 0.45 ± 0.05 |
| 04 | Dajti | 31.5 ± 0.5 | 0.50 ± 0.06 |
| 05 | Linzě | 33.0 ± 0.5 | 0.75 ± 0.05 |
| 06 | Skrapar: Melove | 29.0 ± 1.0 | 0.60 ± 0.02 |
| 07 | Verbas | 35.0 ± 1.0 | 0.28 ± 0.02 |
| 08 | Kuçovë | 23.0 ± 1.0 | 0.20 ± 0.02 |
| 09 | Krujë: Larushk | 28.0 ± 1.0 | 0.30 ± 0.02 |
| 10 | Korçë | 34.0 ± 1.5 | 0.22 ± 0.02 |
| 11 | Poshnje: Berat | 34.0 ± 1.0 | 0.52 ± 0.03 |
| 12 | Karbunarë Lushnjë | 25.0 ± 0.5 | 0.35 ± 0.02 |
| 13 | Karbunarë e Vogël | 38.0 ± 1.5 | 0.37 ± 0.02 |
| 14 | Kuçovë: Rruga S. Zuber | 31.0 ± 1.0 | 0.53 ± 0.02 |
| 15 | Kuçovë: Rruga Kozare | 34.0 ± 1.0 | 0.40 ± 0.02 |
| 16 | Gjokaj | 28.0 ± 0.5 | 0.20 ± 0.02 |
| 17 | Tirana: Botanic Garden | 21.0 ± 0.5 | 0.10 ± 0.01 |
| 18 | Tirana: Grand Park | 27.5 ± 0.5 | 0.15 ± 0.01 |
| 19 | Skrapar: Tomorri Mount. | 29.0 ± 1.0 | 0.40 ± 0.02 |
| 20 | Sheqeras | 29.0 ± 1.0 | 0.20 ± 0.02 |
| 21 | Krujë: Droja | 35.0 ± 1.0 | 0.34 ± 0.02 |
| 22 | Tapizě | 33.0 ± 1.0 | 0.40 ± 0.02 |
| 23 | Kuçovë: Parku i Qytetit | 35.0 ± 1.0 | 0.44 ± 0.03 |
| 24 | Poshnje | 24.0 ± 1.0 | 0.35 ± 0.02 |
| 25 | Lezhě | 36.0 ± 1.5 | 0.36 ± 0.02 |
| 26 | Lezhě Kolsh | 39.0 ± 1.0 | 0.42 ± 0.03 |
| 27 | Fushě Prezě: Palaq | 31.0 ± 1.0 | 0.40 ± 0.02 |
| 28 | Fushě Prezě | 39.0 ± 1.0 | 0.26 ± 0.02 |
| 29 | Fier | 27.0 ± 1.0 | 0.10 ± 0.01 |

*3.2. Qualitative and Quantitative Composition of Essential Oils*

The total chemical profile of the tested essential oils of three selected samples from the localities of Poshnje Berat (locality 11), Krujë Larushk (locality 09), and Lezhë Kolsh (locality 26) are shown in Figure 2, whereas the qualitative and quantitative compositions of the isolated essential oils are shown in Table 3.

However, the quantitative composition of the individual components in the isolated essential oils differed. Essential oil extracted from the chamomile flower sample from the Poshnje Berat locality was recorded to contain 43 chemical components. It was found that /-/-α-bisabololoxide B was the major constituent (45.47 ± 3.5%), followed by /-/-α-bisabolol (17.01 ± 1.0%), cis-, trans-en-yn-dicycloether (10.98 ± 1.0%), /-/-α-bisabololoxide A (10.33 ± 1.0%), and chamazulene (0.63 ± 0.1%). The essential oil composition from Krujë: Larushk had 23 chemical constituents determined, with dominant /-/-α-bisabololoxide A (34.95 ± 3.0%) and /-/-α-bisabololoxide B (26.62 ± 2.0%), followed by /-/-α-bisabolol

(16.27 ± 1.0%), cis-, trans-en-yn-dicycloether (5.16 ± 0.7%), and chamazulene (3.71 ± 0.2%). The content of the most abundant components (/-/-α-bisabolol and chamazulene) was highest in the oil isolated from the Lezhë: Kolsh (26) locality (39.69 ± 2.0% and 6.16 ± 1.0%, respectively; Table 3). In addition, pre-treatment resulted in a higher content of cis-, trans-en-yn-dicycloether (17.24 ± 1.5%), /-/-α-bisabololoxide A (16.78 ± 1.0%), and /-/-α-bisabololoxide B (10.24 ± 1.0%) from 27 identified secondary metabolites.

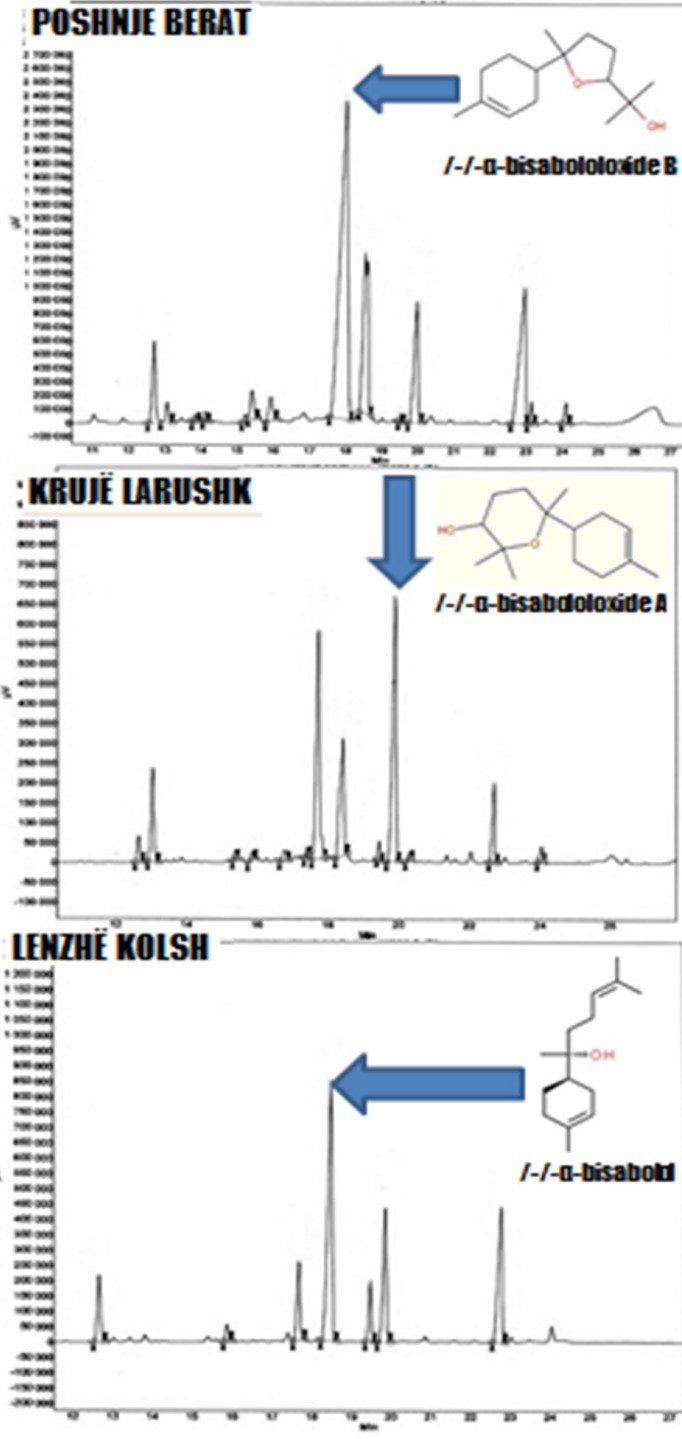

**Figure 2.** The differences in the chromatographic profiles of chamomile essential oil from three Albanian localities: Poshnje Berat, Krujë Larushk and Lezhë Kolsh.

**Table 3.** Main components of the essential oil isolated from chamomile inflorescences belonging to central Albania.

| | Localities | Basic Composition of Essential Oil in % | | | | | |
|---|---|---|---|---|---|---|---|
| | | Fa | Bo | Ch | BoA | BoB | *c*-,*t*-Dc |
| 01 | Baldushk | 3.5 ± 0.5 | 11.5 ± 1.0 | 1.0 ± 0.1 | 12.7 ± 1.0 | 29.5 ± 2.0 | 7.0 ± 1.0 |
| 02 | Rruga Gjiri i Lazit | 4.0 ± 0.5 | 12.5 ± 1.0 | 4.0 ± 0.5 | 20.5 ± 2.5 | 26.0 ± 2.0 | 9.0 ± 1.5 |
| 03 | Zall Her | 8.2 ± 1.5 | 13.5 ± 1.0 | 11.0 ± 0.5 | 11.5 ± 1.5 | 30.5 ± 2.5 | 8.5 ± 1.0 |
| 04 | Dajti | 8.2 ± 1.5 | 15.5 ± 2.0 | 7.5 ± 1.5 | 12.0 ± 1.5 | 32.0 ± 3.0 | 5.5 ± 0.5 |
| 05 | Linzë | 8.2 ± 1.5 | 16.2 ± 2.0 | 5.0 ± 0.7 | 14.0 ± 1.5 | 31.0 ± 3.0 | 5.0 ± 0.5 |
| 06 | Skrapar: Melove | 7.0 ± 1.0 | 16.0 ± 1.0 | 1.5 ± 0.2 | 16.0 ± 1.0 | 28.0 ± 2.0 | 12.0 ± 2.0 |
| 07 | Verbas | 11.0 ± 1.0 | 16.0 ± 1.0 | 1.2 ± 0.2 | 14.1 ± 1.0 | 16.0 ± 1.0 | 16.0 ± 2.5 |
| 08 | Kuçovë | 2.5 ± 0.5 | 16.0 ± 1.0 | 10.0 ± 1.0 | 12.5 ± 0.5 | 29.0 ± 2.0 | 8.0 ± 0.5 |
| 09 | Krujë Larushk | 2.0 ± 0.2 | 16.1 ± 1.0 | 1.6 ± 0.2 | 32.0 ± 3.0 | 24.0 ± 2.0 | 7.5 ± 1.5 |
| 10 | Korçë | 3.0 ± 0.5 | 17.0 ± 1.0 | 0.8 ± 0.1 | 27.0 ± 2.0 | 32.0 ± 3.0 | 16.0 ± 2.0 |
| 11 | Poshnje Berat | 3.0 ± 0.5 | 16.0 ± 1.0 | 0.3 ± 0.1 | 10.0 ± 1.0 | 46.0 ± 3.5 | 12.0 ± 1.5 |
| 12 | Karbunarë Lushnjë | 6.0 ± 1.0 | 16.5 ± 1.0 | 1.5 ± 0.1 | 9.5 ± 1.0 | 26.7 ± 1.5 | 10.0 ± 1.0 |
| 13 | Karbunarë e Vogël | 5.1 ± 1.0 | 18.0 ± 1.0 | 0.7 ± 0.2 | 24.0 ± 2.0 | 26.0 ± 1.5 | 6.0 ± 0.5 |
| 14 | Kuçovë: Rruga S. Zuber | 5.1 ± 1.0 | 19.0 ± 1.0 | 2.0 ± 0.2 | 18.0 ± 1.0 | 29.0 ± 2.0 | 10.0 ± 1.0 |
| 15 | Kuçovë: Rruga Kozare | 4.0 ± 0.5 | 18.0 ± 1.0 | 1.4 ± 0.2 | 22.0 ± 2.0 | 36.0 ± 2.0 | 12.0 ± 2.0 |
| 16 | Gjokaj | 5.7 ± 1.0 | 18.5 ± 1.0 | 0.5 ± 0.1 | 14.2 ± 2.0 | 32.7 ± 2.0 | 9.0 ± 1.5 |
| 17 | Tirana: Botanic Garden | 6.5 ± 1.0 | 24.0 ± 1.0 | 1.5 ± 0.4 | 8.0 ± 1.0 | 19.7 ± 2.0 | 4.5 ± 0.5 |
| 18 | Tirana: Grand Park | 4.5 ± 1.0 | 25.0 ± 3.0 | 1.5 ± 0.4 | 6.0 ± 1.0 | 25.0 ± 3.0 | 5.0 ± 0.5 |
| 19 | Skrapar: Tomorri Mount. | 4.0 ± 1.0 | 20.0 ± 3.0 | 1.0 ± 0.2 | 13.0 ± 1.0 | 33.0 ± 3.0 | 17.0 ± 2.5 |
| 20 | Sheqeras | 7.0 ± 1.0 | 20.0 ± 3.0 | 0.5 ± 0.1 | 20.0 ± 2.0 | 20.0 ± 2.0 | 10.0 ± 1.0 |
| 21 | Krujë: Droja | 9.0 ± 1.0 | 21.0 ± 2.0 | 3.0 ± 0.1 | 16.0 ± 1.0 | 26.0 ± 2.0 | 17.0 ± 2.0 |
| 22 | Tapizë | 6.0 ± 1.0 | 22.0 ± 2.0 | 1.5 ± 0.2 | 15.0 ± 1.0 | 32.0 ± 2.0 | 15.0 ± 2.0 |
| 23 | Kuçovë: Parku i Qytetit | 13.0 ± 1.0 | 24.0 ± 2.0 | 1.0 ± 0.2 | 18.0 ± 2.0 | 27.0 ± 2.0 | 15.0 ± 1.5 |
| 24 | Poshnje | 6.0 ± 1.0 | 30.0 ± 2.0 | 1.2 ± 0.2 | 6.0 ± 1.0 | 37.0 ± 2.0 | 7.0 ± 1.0 |
| 25 | Lezhë | 4.0 ± 0.5 | 36.0 ± 2.0 | 5.0 ± 0.2 | 22.0 ± 2.0 | 19.0 ± 1.0 | 16.0 ± 2.0 |
| 26 | Lezhë Kolsh | 7.0 ± 0.5 | 42.0 ± 2.0 | 6.0 ± 1.0 | 11.0 ± 1.0 | 18.0 ± 1.0 | 19.0 ± 2.5 |
| 27 | Fushë Prezë: Palaq | 5.0 ± 1.0 | 42.0 ± 2.0 | 0.4 ± 1.0 | 4.0 ± 1.0 | 20.0 ± 2.0 | 14.0 ± 1.0 |
| 28 | Fushë Prezë | 9.0 ± 1.0 | 40.0 ± 2.0 | 1.0 ± 1.0 | 10.0 ± 1.0 | 23.0 ± 2.0 | 9.0 ± 1.0 |
| 29 | Fier | 4.0 ± 1.0 | 42.0 ± 2.0 | 1.0 ± 1.0 | 11.0 ± 1.0 | 20.0 ± 2.0 | 12.0 ± 1.5 |

Fa—trans-β-farnesene; Bo—/-/-α-bisabolol; Ch—chamazulene; BoA—/-/-α-bisabololoxide A; BoB—/-/-α-bisabololoxide B; *c*-,*t*-Dc—cis-, trans-en-yn-dicycloether.

### 3.3. Chamomile Chemical Types

　　　The qualitative–quantitative essential oil diversity of wild-growing chamomile populations is shown in Table 3. The frequency of the population of /-/-α-bisabololoxide B content is very high (from 16.0 ± 1.0 to 45.47 ± 3.5%) in the regions located at Baldush (locality 01), Rruga Gjirit i Lazit (locality 02), Zall Her (locality 03), Dajti (locality 04), Linzë (locality 05), Skrapar: Melove (locality 06), Verbas (locality 07), Kucovë (locality 08), Krujë: Larushk (locality 09), Korçë (locality 10), Poshnje: Berat (locality 11), Karbunarë Lushnjë (locality 12), Karbunarë e Vogël (locality 13), Kuçovë: Rruga S. Zuber (locality 14), Kuçovë: Rruga Kozare (locality 15), Gjokaj (locality 16), Tirana: Botanic Garden and Grand Park (localities 17, 18), Skrapar: Tomorri Mountains (locality 19), Sheqeras (locality 20), Krujë:

Droja (locality 21), Tapizë (22), Kuçovë: Parku i Qytetit (locality 23), and Poshnje (locality 24). The chamomile plants with a high /-/-α- bisabolol content were very rare. The range of this oxide was from 6.0 ± 1.0 to 32.0± 3.0%. The frequency of the high /-/-α-bisabolol content (from 40.0 ± 2.0 to 42.0 ± 2.0%) was in the localities of Lezhë: Kolsh (locality 26), Fier (locality 29), and Fushë Prezë (localities 27, 28). However, the accumulations of chamazulene content (from 1.0 ± 0.1 to 11.0 ± 0.5%) were affected by the conditions of the year, and regional differences were established.

As a result, four basic types of chamomile A, B, C, and D are recognized, according to the qualitative and quantitative composition of the essential oils (Table 4 and Figure 2), which are characterized as follows: type A (/-/-α-bisabololoxide A is the dominant component), type B ( /-/-α-bisabololoxide B is the dominant component), type C (typically with the highest /-/-α-bisabolol content), and type D (/-/-α-bisabolol and /-/-α-bisabolol oxide A and B present in 1:1 ratio approximately).

**Table 4.** The marked differences in chemical composition of chamomile essential oil from tree localities in central Albania [%].

| Component as % of Essential Oil * | GC-MS $t_R$ (min.) ** | Kovat's Index | Localities of Chamomile Flower Collection | | |
|---|---|---|---|---|---|
| | | | Poshnje Berat | Krujë Larushk | Lezhë Kolsh |
| trans-2-hexanal | 9.26 | 842 | 1.15 | 1.06 | —— |
| tricyclene | 10.41 | 879 | 0.03 | 0.96 | —— |
| α-pinene | 14.81 | 929 | 0.03 | 0.19 | —— |
| camphene | 15.23 | 942 | 0.02 | 0.36 | —— |
| sabinene | 15.53 | 973 | 0.02 | —— | —— |
| β-pinene | 15.85 | 984 | 0.03 | —— | —— |
| β-myrcene | 16.15 | 989 | 0.09 | —— | —— |
| p-cymene | 16.55 | 1032 | 0.04 | —— | —— |
| (ε,ε)-matricaria esther | 19.07 | 1491 | 1.27 | 1.63 | 0.60 |
| γ-muurolene | 19.24 | 1508 | 0.07 | —— | —— |
| β-farnesene | 19.55 | 1513 | 3.09 | 1.15 | 2.63 |
| α-muurolene | 19.89 | 1521 | 0.25 | —— | —— |
| α-bulnesene | 20.05 | 1528 | 0.11 | —— | 0.06 |
| γ-cadinene | 20.23 | 1539 | 0.09 | —— | 0.22 |
| calamenene | 20.65 | 1543 | 0.07 | —— | 0.02 |
| α-acoradiene | 20.81 | 1554 | 0.13 | —— | —— |
| cadina-1,4-diene | 20.99 | 1558 | 0.08 | —— | —— |
| δ-cadinene | 23.22 | 1563 | 0.02 | —— | —— |
| α-amorphen | 23.49 | 1564 | 0.05 | —— | —— |
| α-calacorene | 24.07 | 1567 | 0.09 | —— | —— |
| trans-nerolidol | 24.89 | 1573 | 0.77 | —— | —— |
| epiglobulol | 25.48 | 1584 | 0.52 | —— | 0.15 |
| junenol | 26.07 | 1590 | 0.09 | —— | 0.06 |
| spatulenol | 26.79 | 1602 | 0.15 | 0.72 | 0.08 |
| β-eudesnol | 26.97 | 1603 | 0.10 | —— | 0.21 |
| globulol | 27.15 | 1611 | 0.06 | —— | 0.11 |
| tremetone | 27.34 | 1615 | 0.33 | —— | 0.34 |

**Table 4.** *Cont.*

| Component as % of Essential Oil * | GC-MS t$_R$ (min.) ** | Kovat's Index | Localities of Chamomile Flower Collection | | |
|---|---|---|---|---|---|
| | | | Poshnje Berat | Krujë Larushk | Lezhë Kolsh |
| caryophyllene oxide | 27.9 | 1617 | 0.04 | 0.02 | 0.06 |
| viridiflorene | 28.06 | 1626 | 0.48 | 0.18 | 0.08 |
| dillapiole | 28.52 | 1641 | 0.72 | 0.77 | 0.19 |
| cubebol | 29.4 | 1652 | 0.38 | 0.25 | 0.10 |
| β-bisabolole | 29.85 | 1660 | 0.30 | 0.28 | 1.01 |
| τ-muurolol | 30.61 | 1663 | 1.64 | 0.51 | 0.99 |
| *α*-bisabololoxide B | 31.28 | 1679 | **45.47** | 26.62 | 10.24 |
| *α*-bisabolonoxide A | 31.69 | 1684 | 0.26 | 0.29 | 1.08 |
| valerianol | 32.05 | 1689 | 2.20 | 2.13 | 0.05 |
| cadalene | 33.05 | 1702 | 0.12 | 0.70 | 0.06 |
| *α*-bisabolol | 33.59 | 1711 | 17.11 | 16.27 | **39.69** |
| chamazulene | 43.4 | 1773 | 0.63 | 3.71 | 6.16 |
| *α*-bisabololoxide A | 44.98 | 1781 | 10.33 | **34.95** | 16.78 |
| guaiazulene | 46.13 | 1802 | 0.18 | 1.09 | 1.91 |
| cis-en-yn-dicycloether | 59.00 | 1927 | 10.82 | 5.02 | 17.18 |
| trans-en-yn-dicycloether | 59.03 | 1942 | 0.16 | 0.14 | 0.06 |
| **Total [%]** | | | 100.00 | 100.00 | 100.00 |

Note: *—data are expressed as area in % of the 100.00 % of all identified peaks; **—retention times.

Analyzing the chamomile samples collected from 29 regions in Albania, 2 main groups were separated: populations accumulating /-/-α-bisabololoxide B, and populations characterized by the presence of /-/-α-bisabolol.

### 3.4. Hierarchical Cluster Analysis

In order to provide additional insights into the chemotypes of chamomile essential oils, we carried out a hierarchical cluster analysis based on the constituents. The dendrogram of the analysis is shown in Figure 3. Based on this analysis, there are two different confirmed chemotypes: chemotype B, dominated by /-/-α-bisabololoxide B and chemotype C, dominated by /-/-α-bisabolol.

The average Euclidean distance among the populations calculated based on the essential oil constituents was 41.6, ranging from 8.9 (between P3 and P4) to 79.1 (between P1 and P5) (Figure 3).

The results obtained from the cluster analysis show the existence of a high variability within the essential oils and differences among the following chamomile populations collected in different areas in Albania (Figure 3). From the 29 populations submitted to multivariate analysis, two well-defined groups of essential oils were differentiated using cluster analysis. Two subclusters can be observed: the first subset contains five populations collected in Fer, Fishe Preze, Palaq, Fushe Preze, Kolsh: Lezhe, and Lezhe, and the second subset includes the samples collected in the other areas.

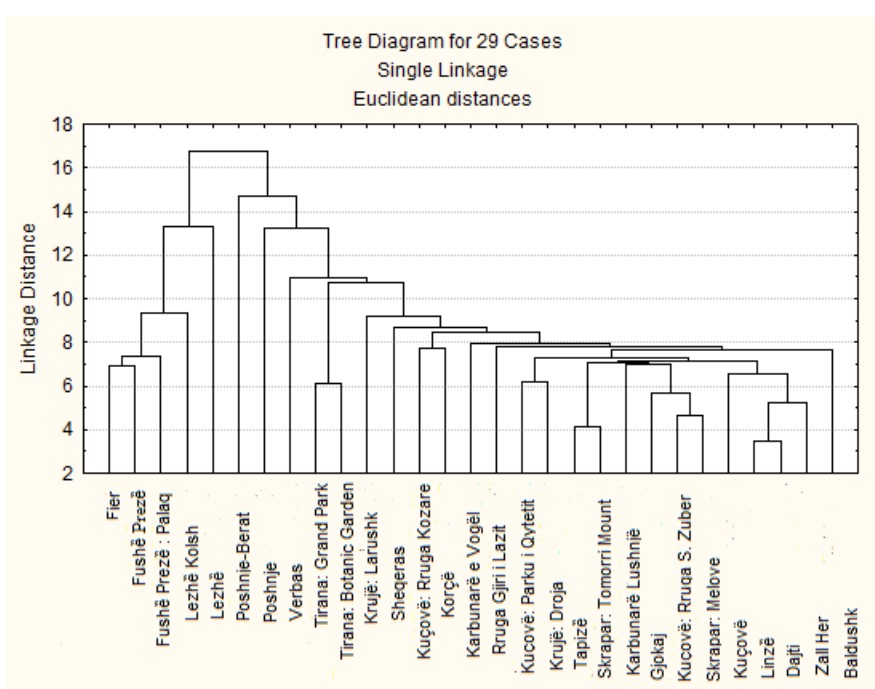

**Figure 3.** Dendrogram of relationships among Albanian chamomile populations and qualitative–quantitative characteristics of essential oils.

## 4. Discussion

The chamomile inflorescences of diploid plants usually contain from 0.4 to 1.0% essential oils. While the quantity of essential oil in the flowers from different localities may be genetically coded, the differences in production at the 29 test sites indicated considerable interaction between the plants and the environment [13]. These observations confirm previous studies in Eastern Slovakia [14] and Iran [2]. In both of these later cases, the range in the essential oil content from 0.4 to 0.8% occurred but was less than the accepted 0.9 to 1.0% variation accepted for chamomile flower drugs of variety certifications.

In a natural ecosystem, chamomile species separated on the bases of their secondary products contribute to the development of the modification of the existing ecological balance [15]. However, for a full understanding of the biosynthetic processes of these plant populations, a detailed analysis of the geographical occurrences and their chemotypes are required.

The compounds of essential oils are clearly genetically determined to a higher degree than their contents. The oil content is more strongly influenced by environmental factors and shows considerable variation, even within a relatively small area [14].

In 1972 and 1973, Schilcher [16,17] found that according to the oil composition, chamomile could be differentiated into chemical types based on the contents of /-/-α-bisabololoxide A and B, /-/-α-bisabolol (type A: /-/-α-bisabololoxide B > /-/-α-bisabololoxide B > /-/-α-bisabolol; type B: /-/-α-bisabololoxide A > /-/-α-bisabololoxide B > /-/-α-bisabolol; type C: /-/-α-bisabolol > /-/-α-bisabololoxide B > /-/-α-bisabololoxide A; and type D: /-/-α-bisabololoxide B ≈ /-/-α-bisabololoxide A ≈ /-/-α-bisabolol).

With regard to the sesquiterpene alcohols, the wild plants of this species mostly show bisabololoxides. The plant populations rich in /-/α- bisabolol could be found endemically in Catalonia (Spain) and, to a smaller extent, in other populations. This fact was confirmed by the study of small populations on the islands of Malta, Cyprus, and Crimea [18].

Analyses of the essential oils obtained from Hungarian, Czech, Slovak, German, and Spanish-grown chamomile flowers by Honcariv and Repcak [19] revealed that the chemical composition of chamomile oil varied extremely widely. They fully confirmed the existence of chemotypes in the chamomile species.

The chemical diversity of chamomile populations was justified in the Eastern Slovakian Lowland as well. It was proved by the investigations of Salamon (2004) [20] analyzing populations for the presence of essential oil and its composition during the period of 1995–1998.

In the chamomile samples from Attica in Greece, after essential oil hydro-distillation and GC/MS identification, 21 components were determined in 1993. The major constituents were β-farnesene (21.2 and 5.6%) and /-/-α-bisabololoxide A (16.0 and 9.0%) [21].

In Italy [22], the compounds of the oil from tubular and ligulae florets and from the receptacle were (E)-β-farnesene (14.4–17.1%), spathulenol (4.4–12.6%), α-bisabolone oxide A (9.2–11.2%), chamazulene (8.4–13.7%), α-bisabolol oxide A (4.9–11.6%), and cis-en-yn-bicycloether (2.7–13.4%). The bisabololoxide chemotype was proven.

The chemotype C chamomile populations were located in the northern, southwestern and southern parts of Bulgaria. The ones from the northwestern and southwestern regions had an average of 39% /-/-α-bisabolol and blue oil with 2.7% chamazulene content. Chamomile populations with the chemotype A essential oil were found in the central regions of North Bulgaria. They had an average of 34% /-/-α-bisabololoxide A and blue essential oil with the highest chamazulene content of 7.7% for all chemotypes. Chemotype D chamomile, with almost equal amounts of (-)-α-bisabolol and (-)-bisabolol oxide A and B, was discovered on one single site near to the city of Kjustendil [23].

The essential oil composition of the Hungarian populations can be divided into two groups: the α-bisabolol chemotype (where the ratio of this component is between 45 and 58%) and the bisabololoxide A chemotype (where the ratio of bisabololoxide A varied between 34 and 43%) [24].

In Poland [25], the presence of 22 compounds was identified in the essential oil obtained from the inflorescences of wild-growing chamomile, among which sesquiterpene compounds had the highest proportions: the dominant compounds were /-/-α-bisabololoxide A (31.70%), /-/-α-bisabololoxide B (17.09%), and /-/-α-bisabolnoxide A (15.73%); and chamazulene was also found in a large amount (15.58%). Among the other compounds, the presence of β-farnesene was found in small quantities (4.89%).

In total, 13 Brazilian chamomile populations from different geographic origins, agricultural practices, and harvest times, grown in the Santa Catarina and Paraná states in Brazil were analyzed [26]. Eleven samples were classified as chemotype B (rich in /-/-α-bisabololoxide B), and the remaining two were classified as chemotype A (rich in /-/-α-bisabololoxide A). The predominance of chemotype B was observed concerning the agricultural practices (organic or conventional) or geographic origin (Santa Catarina or Paraná).

Finally, there is the latest study of chamomile chemical profiles from plants collected from semi-wild populations in Latvia [27]. All samples were characterized with a high quality, reaching the thresholds of European Pharmacopoeia for essential oil content with high levels of /-/-α-bisabololoxide A.

This scientific study of the chamomile populations was complemented by a research on the variability in Albania, focusing on morphological traits [27,28]. To characterize and identify the different chemotypes of the Albanian populations and their essential oil composition using cluster analysis, it is possible to compare with different results and the dendrogram of the *Matricaria recutita* populations in Iran [29].

Thus, chamomile with a particular chemical composition is used as a drug as it shows specific pharmacological activity. As efficient methods for determining the drug's constituents and effectiveness have been developed, the content of /-/-α-bisabolol and its oxides in the flowers has become an important indicator of the raw material's quality and value.

## 5. Conclusions

The cultivation of chamomile is practiced all over the world; however, the collection of the wild population has not lost its importance, especially in European countries in

the east and south. In Albania, the blossom of wild chamomile starts in the second half of May and continues until the beginning of July. Much of the Albanian production of chamomile is harvested from wild or unselected populations and is hand-harvested by low-cost labor. The essential oil contents from the localities of Zall Her, Dajti, Kuçovë, and Linzë had a deeper blue color. The yield ranged from $0.10 \pm 0.01\%$ to $0.75 \pm 0.05\%$ on a dry-weight basis, which is an insufficient amount. Analyses of the essential oil obtained from Albanian wild-grown chamomile reveal that the chemical composition of the oil varied extremely widely with the dominant chemotype B. As a result, four basic chemical types (A, B, C, and D) were recognized, according to the quality of the product. The country, as the major supplier of this raw material for the world market, needs to initial intensive improvement programs to produce chamomile with high levels of oils with a defined chemical composition placing significance on /-/-α-bisabolol and chamazulene.

**Author Contributions:** I.S. processed the experimental data, performed the biochemical–biological analysis, and drafted the manuscript. A.I. carried out almost all of the technical details and performed numerical calculations for the suggested experiment with plant population's collection in Albania. M.K. devised the project of Thymus biodiversity, the main conceptual ideas, and proof outline. All authors discussed the results and commented on the manuscript. All authors have read and agreed to the published version of the manuscript.

**Funding:** This research received no external funding.

**Institutional Review Board Statement:** Not applicable for studies not involving humans or animals.

**Informed Consent Statement:** Not applicable for studies not involving humans.

**Data Availability Statement:** The data are openly available in a public repository that issues datasets with DOIs.

**Conflicts of Interest:** The authors declare that the research was conducted in the absence of any commercial or financial relationships that could be construed as a potential conflict of interest.

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
