# Peer review of "Essential Oil Content and Composition of the Chamomile Inflorescences (Matricaria recutita L.) Belonging to Central Albania"

_horticulturae, doi:10.3390/horticulturae9010047_

Round 1
Reviewer 1 Report
The authors of the paper present the results of essential oils content and their sesquiterpene compositions of 29 Albanian autochthonous populations of chamomile. Chamomile (Matricaria recutita) is а well known medicinal and cosmetic plant and study of local genetic resources is important for development of its breeding. The oil components were analyzed by Vega Series Carloerba Gas Chromatograph, statistical analysis was applied. The data obtained are clear presented in the tables and figures. The authors know well the publications on this issue that is shown in the discussion section.
There are some comments to the manuscript:
1) Abstract: According to the classification described by the authors in section 3.3. the chemical type of studied chamomile populations is mostly B, but not A
2) Section 3.1. The phrase “Chamomile inflorescences usually contain 0.1 – 1.5 % (tetraploid plants), usually 0.4 – 0.9 %, volatile oil” is not correct
3) Section 3.3 (line 9) – it should be /-/-α-bisaboloxid A
4) There is no reference to the Table 4.
5) Section 3.4. The authors refer to the principal component analysis, but it is not shown. So, the reference should be deleted, or PCA should be done
Author Response
Dear reviewer,
thank you very much for reviewing my manuscript - an original scientific paper on a large-scale study of chamomile populations in Albania. I appreciate your knowledge and pointing out the shortcomings in the text of the manuscript.
On the basis of noted errors and shortcomings - I revised the manuscript, in which I tried to take into account your comments and observations.
I am asking for his subsequent review and comment on the publication of my contribution.
Thank you for your cooperation.

Reviewer 2 Report
The manuscript title “Essential Oil Content and its Composition of the Chamomile Inflorescences (Matricaria recutita L.) belonging to the Central Part in Albania” is good study but needs significant improvements. I have some comments and suggestions for authors:
Comments for authors:
1- Please give a conclusion of study in the (2-3 lines) in the end of abstract.
2- The first three paragraphs of introduction section have no citation/reference. Please remove all the un-cited statements from the MS or provide appropriate citations in the MS.
3- Which compounds have been already (from previous published studies) identified in the Matricaria recutita, should be added in the introduction section as its background. Currently, only three references have been cited in the introduction section….
4- Figure 2 legends should be placed below the figure 2, in the MS. Same issue with figure 3.
5- Figure 2 also seems blurring, if authors have a better resolution than please provide a better copy of figure 2.
6- The conclusion section first 7 lines are not conclusion of this study, it seems discussion, I suggest authors delete these lines and add conclusion/ major findings/outcomes and future research direction in the conclusion.
7- The last reference sequence number in wrong, please correct this.
Author Response

(The authors gave the same response as above.)

Reviewer 3 Report
Refer to the attached reviewer-annotated manuscript for scientific and editorial issues that needs attention.
Grammar poor - needs professional editing
Abstract/Materials and Methods/Results and Discussion: Was a correlation analysis/Chi-square analysis done to determine if the influences of habitat differences correlated with volatile oil?
Abstract - Add: It is concluded that ...
Introduction:
Significant amounts of redundant information need to be deleted - refer to the reviewer's annotated manuscript.
Paragraph 1: No sources cited - please add
Restructure the discussion by using the same sub-headings as were used in the results section. This is to: (1) align the two sections; (2) make orientation for the reader easier; (3) preserve the sequence of reporting; (4) preserve the weightings between results and discussion (many results = lots of discussion, and vice versa)
1.1. The Dependence of Essential Chamomile Oil Yield from their Habitats
1.2. Qualitative and Quantitative Composition of Essential Oil
1.3. Chamomile Chemical Types
1.4. Hierarchical Cluster Analysis
Conclusion:
This is NOT a Summary! INSIGHT NEEDED!
Replace with (1) what is new to the study; (2) what are the significance of the new findings; (3) what are the gaps in the investigation; (4) How to close the gaps; (5) What is the holistic conclusion; (6) What is the holistic recommendation (way forward)

Author Response
Dear reviewer,
first of all, I would like to thank you for your comments and improvements to my text of the original scientific work, which deals with the extensive study of chamomile populations and the determination of their chemotypes in Albania. Thank you very much for the corrections of the English language that you made in the text.
I admit, as a scientist coming from a country where we do not speak or write English - mistakes and stylistic inconsistencies may occur in such a presentation. I appreciate people taking this into account and helping us deal with it. This is also your case, so thank you very much.
I completely reworked my manuscript based on your notes, comments and recommendations. I believe that the current manuscript meets the quality you require.
However, if there are other inconsistencies, please mark them in the text, I will be happy to make further adjustments and additions.
Sincerely, Author
Dear reviewer,
first of all, I would like to thank you for your comments and improvements to my text of the original scientific work, which deals with the extensive study of chamomile populations and the determination of their chemotypes in Albania. Thank you very much for the corrections of the English language that you made in the text.
I admit, as a scientist coming from a country where we do not speak or write English - mistakes and stylistic inconsistencies may occur in such a presentation. I appreciate people taking this into account and helping us deal with it. This is also your case, so thank you very much.
I completely reworked my manuscript based on your notes, comments and recommendations. I believe that the current manuscript meets the quality you require.
However, if there are other inconsistencies, please mark them in the text, I will be happy to make further adjustments and additions.
Sincerely, Author
